# The Significance of Tablet Internal Structure on Disintegration and Dissolution of Immediate-Release Formulas: A Review

**Camila G. Jange [1], Carl R. Wassgren [2,3]**  **and Kingsly Ambrose [1,***

[1]   Department of Agricultural and Biological Engineering, Purdue University, West Lafayette, IN 47907, USA
[2]   School of Mechanical Engineering, Purdue University, West Lafayette, IN 47907, USA
[3]   Department of Industrial and Physical Pharmacy (By Courtesy), Purdue University, West Lafayette, IN 47907, USA
[*]   Correspondence: rambrose@purdue.edu; Tel.: +1-765-494-6599

**Abstract:** The internal microstructure of a tablet, such as pore geometry and pore volume, impacts the tablet's disintegration kinetics. Ideally, one could design the microstructure to control dissolution onset and therapeutical performance of immediate-release formulas; however, manufacturing tablets with a desired microstructure can be challenging due to the interplay between formulation and process parameters. Direct quantification of tablet microstructure can provide a framework for optimizing composition and process parameters based on a Quality-by-Design approach. This article reviews the importance of tablet microstructure design and liquid transport kinetics to help optimize the release and dissolution profiles of immediate-release products. Additionally, the formulation and process parameters influencing the tablet microstructure and liquid transport kinetics are discussed.

**Keywords:** pore structure; excipient functionality; disintegration; dissolution

## 1. Introduction

Solid dosage forms, such as tablets, are important delivery systems for the oral administration of active pharmaceutical ingredients (APIs). The use of solid dosage forms for therapeutical treatment has some advantages due to their portability and compactness, noninvasive and self-administering capabilities, and promptness of production in discrete units. These characteristics facilitate dosing and are relatively cheaper to manufacture compared with liquid and parenteral formulas. Among the solid dosage forms, tablets have advantages over capsules due to their longer shelf life, higher dosing capacity, and tunable release mechanisms. For instance, tablets are generally designed to either be immediate- or modified-release products. An immediate-release product is used when rapid dissolution is required, while a modified-release product has gradual and sustained dissolution for selective absorption or a controlled release response. The release and dissolution of immediate-release formulas are generally more affected by the microstructure design that is dependent on the interplay between the formulation and process design.

Immediate-release tablets break into smaller particles through a process referred to as disintegration. Some limitations in developing direct techniques for evaluating disintegration profiles are linked to drawbacks in the disintegration testing apparatus and lack of theoretical models, difficulties in distinguishing the interplay between process and formulation parameters, and the complexity of tablet microstructure dynamics in liquids [1]. Often, the disintegration kinetics is not the rate-limiting process for complete drug dissolution but rather the limiting factor for drug liberation.

In general, in vitro experiments are used to estimate the in vivo performance of immediate-release tablets [2]. In addition to characterizing the release response, a dissolution test can be used to evaluate the variability in formulation and tablet properties. Identifying the root causes of variability is difficult due to the destructive nature of in vitro studies [2,3]. Additionally, in vitro trials do not differentiate between disintegration and

dissolution. Semi-empirical models have been proposed to predict dissolution behavior; however, these models possess important limitations. For instance, a universal semi-empirical model is difficult to establish due to variability in mass transfer mechanisms [4]. Therefore, the translation of tablet properties from the bulk (i.e., friability and tablet breaking force) to the micro-level becomes paramount for controlling and troubleshooting problems in tablet manufacturing. It is known that a tablet's in vitro performance can be controlled by the tablet's formulation and structure, the latter being determined by processing conditions. The selection of process and formulation variables can help to tune the tablet's pore structure, which controls the internal liquid transport and dissolution behavior [5,6]. To improve the control of liquid transport through tablet matrices, the Quality-by-Design (QbD) approach has gained momentum since it offers a framework that uses non-destructive techniques and statistical and machine learning to predict the disintegration and dissolution of tablets.

This literature review focuses on the fundamentals of liquid diffusion into a solid substrate and its importance in understanding the disintegration and dissolution of tablets. In addition, mathematical models that describe capillary transport are presented. Furthermore, a general overview of disintegration and dissolution kinetics are discussed in this review. A discussion on emerging technologies to assess internal microstructure and decouple disintegration and dissolution are also included in this review. The final section of this article presents an overview of the influence of formulation and process parameters on tablet microstructure and how it affects disintegration and dissolution kinetics. The final section also includes examples of the Quality-by-Design (QbD) approach in the design of immediate-release products.

## 2. Liquid Transport Kinetics

Tablet disintegration is the destruction of particle bonds to produce fragments, small agglomerates, or primary particles [7]. The interaction between a liquid and a submerged porous structure is composed of kinematic (rapid wetting) and capillary (fluid penetration) stages. More specifically, the disintegration mechanism of a compact involves rapid spreading of liquid, liquid penetration, strain recovery, and swelling [8]. The section on the mechanisms of disintegration outlines the detailed aspects of the rate processes during the breakdown of immediate-release dosage forms. Liquid penetration into the internal voids may cause instantaneous bond breakage or trigger relaxational mechanisms, such as unidirectional or omni-directional tablet swelling. Relaxation leads to interparticle bond disruption if the cohesive stresses within the tablet structure are smaller than the loading stresses generated during swelling. It is well known that the pore volume, tortuosity (related to pore connectivity, shape, size, and orientation), and compact density can limit the mass transfer of liquid through the pore space [9]. Formulation design also plays a crucial role in the evolution of a submerged tablet's microstructure. The effect of formulation design is presented later in this review.

Rapid spreading and penetration are difficult to distinguish during dissolution testing because of the overlap in their time scales. Spreading, which takes place on the order of milliseconds/seconds, occurs first and is dependent on the fluid properties (density, viscosity, surface tension, wettability) and solid properties (surface roughness) [10,11]. Absorption is limited by the liquid properties and tablet's microstructural properties, such as wettability and pore size, respectively [10,11]. Therefore, given the direct relation between fluid penetration and microstructure characteristics, the following section is focused on liquid penetration into porous substrates (capillary transport).

### 2.1. Vertical Liquid Transport

The liquid penetration theory includes the stages of (i) penetration normal to the surface into the capillaries, (ii) radial spreading far from the point of entry, and (iii) spreading tangentially across the porous layers. Based on the Washburn law of imbibition, a generalized model can be used to predict the first stage of liquid penetration [12,13]. Us-

ing this model, the pores of a substrate can be approximated as capillary tubes with the penetration length for each capillary being the sum of each pore volume divided by the corresponding pore surface area. The Washburn law serves as a practical analytical tool to differentiate the initial penetration phenomenon in different porous structures [14–16]. Model discrepancies are expected when the porosity distribution is bimodal, or the contact angles exhibit dynamic changes during liquid penetration. In addition, tortuosity and pore anisotropy can deviate from the cylindrical capillary approximation [12,17–19].

For the first stage of liquid penetration, an experiment developed by White (cited in [17]) is discussed to explain the permeability of liquids in vertical flow. In the White experiment, a packed powder bed was first saturated with liquid A or air before contact with an immiscible liquid B. The capillary rise was ceased by external pressure ($P_e$). The capillary force initiates liquid penetration until reaching an equilibrium with the external and gravitational forces (Equation (1)):

$$(\sigma_{SV} - \sigma_{SL})\rho_S A_{sp}\varnothing_S \pi R_t^2 = \rho_L g V_L + P_e(1 - \varnothing_S)\pi R_t^2 \tag{1}$$

where $\sigma_{SV}$ is the surface tension of a liquid at the solid–vapor interface, $\sigma_{SL}$ is the surface tension of a liquid at the solid–liquid interface, $\rho_S$ is the solid's density, $A_{sp}$ is the surface area of the pore, $\varnothing_S$ is the solid fraction, $R_t$ is the radius of the solid material (measured from the bounder bed lateral until the capillary), $\rho_L$ is the density of liquid B, $V_L$ is the liquid B volume filling the empty capillaries at equilibrium, and g is the gravitational acceleration.

The liquid volume can be calculated as Equation (2):

$$V_L = (1 - \varnothing_S)\pi R_t^2 H_{eq} \tag{2}$$

where $H_{eq}$ is the capillary length at equilibrium (considering liquid filling the empty spaces between particles at equilibrium).

Rewriting Equation (1) in terms of Equation (3):

$$(\sigma_{SV} - \sigma_{SL})\rho_S A_{sp}\varnothing_S \pi R_t^2 = \rho_L g(1 - \varnothing_S)\pi R_t^2 H_{eq} + P_e(1 - \varnothing_S)\pi R_t^2 \tag{3}$$

Rewriting Equation (3) in terms of external pressure (Equation (4)):

$$P_e = \frac{(\sigma_{SV} - \sigma_{SL})\rho_S A_{sp}\varnothing_S}{(1 - \varnothing_S)} - \rho_L g H_{eq} \tag{4}$$

White (cited in [17]) introduced an effective pore radius, $R_{eff}$ (Equation (5)), which is equivalent to the pore radius, $R_p$.

$$R_{eff} = R_p = \frac{2(1 - \varnothing_S)}{\rho_S A_{sp}\varnothing_S} = \frac{2\varepsilon_V}{\rho_S A_{sp}(1 - \varepsilon_V)} \tag{5}$$

where $\varnothing_S$ is the volume fraction of a particle, $A_{sp}$ is the specific surface area, and $R_p$ is the mean pore radius.

Rewriting the external pressure by neglecting the hydrostatic pressure yields a modified Laplace pressure ($P_C$) as in Equation (6):

$$P_e = P_c = \frac{2\gamma_{LV}}{R_{eff}}\cos\theta_{PL} \tag{6}$$

The liquid spreading in vertical capillaries at the initial stage of liquid penetration can be approximated to a linear flow. This linear flow is a Washburn type of one-dimensional flux. The flux through the capillaries can be expressed by the Poiseuille mean flow rate ($v_L$)

(Equation (7)). This relation is the ratio between the capillary permeability to the pressure gradient ($\Delta P$) over the capillary length ($H_t$) and fluid dynamic viscosity ($\mu$).

$$J_P = \frac{dV/dt}{A_\varnothing} \frac{dN_P \pi R_P^2 H/N_P \pi R_P^2}{dt} = \frac{dH}{dt} = v_L \approx \frac{D_{KC}}{\eta_L}\left(\frac{\Delta P}{H_t}\right) \tag{7}$$

where $A_\varnothing$ is the pore area.

The pore area ($A_\varnothing$) can be calculated by introducing a permeability constant ($D_{KC}$) determined using the Kozeny–Carman relation, which introduces a pore width and a tortuosity factor to incorporate pore heterogeneity effects. The permeability constant is defined as Equation (8). The plot of flux over pressure yields the permeability as the slope of the curve. Equation (8) introduces the functions that represent variations in pore width ($f_p$) and tortuosity ($f_t$). The product of $f_p$ by $f_t$ for random pore systems is known to be $2 \times 2.5 = 5$.

$$D_{KC} = \frac{\varepsilon_V R_H^2}{f_p f_t} \approx \frac{\varepsilon_V R_P^2}{4 f_p f_t} \approx \frac{\varepsilon_V R_P^2}{20} \tag{8}$$

where $\varepsilon_V$ is the bulk porosity, $R_H$ is the effective radius of pores, and $R_p$ is the mean pore radius.

For a cylindrical capillary, the $R_H$ radius is calculated as the ratio between void volume ($V_p$) and pore area ($A_p$) (Equation (9)):

$$R_H = \frac{V_P}{A_P} = \frac{\pi R_P^2 L_P}{2\pi R_P L_P} = \frac{R_P}{2} = \frac{\varepsilon_V}{\rho_S A_{sp}(1-\varepsilon_V)} \tag{9}$$

where $L_p$ is the length of the pore channel (considered as a cylinder).

The Laplace capillary pressure and the permeability factor are then introduced in Equation (7), yielding Equation (10):

$$J_P = \frac{dH}{dt} \approx \frac{\varepsilon_V R_P \gamma_{LV}}{10 \eta_L H_t}\cos\theta_{PL} \tag{10}$$

where the gravitational effect is neglected and $R_p = 2 R_H$, $\gamma_{LV}$ is the liquid surface tension, $\theta_{PL}$ is the contact angle between pore and liquid, $\eta_L$ is the apparent liquid viscosity, and $H_t$ is the capillary length over time.

Integrating Equation (10) yields a version of a modified Washburn Equation (11):

$$H_t = \sqrt{\frac{\varepsilon_V R_P \gamma_{LV}}{5\eta_L}\cos\theta_{PL}}\sqrt{t} \tag{11}$$

where $\varepsilon_V$ is the bulk porosity, $R_p$ the hydrodynamic radii, $\gamma_{LV}$ is the surface tension of the liquid, and $\theta_{PL}$ is the contact angle.

### 2.2. Radial Liquid Transport

During the first stage of transport, the fluid does not radially penetrate. In the second and third stages, the fluid is completely absorbed into the pore channels due to the reduction in overpressure generated by capillary pressure. This absorption front prompts a redistribution of the penetrating fluid from large to small pores. No generalized models are available in the literature for the stages ii and iii of fluid penetration. Instead, power-law or log-log empirical fits are used to relate the time-dependent penetration length to pore area or pore radius [18]. These fits rely on experimental techniques such as magnetic resonance imaging (MRI), neutron-radiography, near-infrared spectroscopy (NIR), and terahertz time-domain spectroscopy (THz-TDS), which can be used to visualize the liquid distribution within a compact. Various experimental methods that have been used to differentiate capillary radius, shape, and tortuosity path length, and their distributions are presented in Section 3.1.

In the radial liquid penetration (stage ii), Gillespie [18] determined the penetration length by considering partial filling of pores. The flow is two-dimensional between two parallel plates at a distance $H_P$. The liquid flow is expressed in terms of the Darcy flow (Poiseuille) equation as in Equation (12):

$$J_R = \frac{dV/dt}{A_\varnothing} = \frac{d\left(\frac{\pi R^2 H_P}{2\pi R H_P}\right)}{dt} = \frac{1}{2}\langle\frac{dR}{dt}\rangle = \frac{\varepsilon_V H_P^2}{12\eta_L}\left(\frac{\Delta P}{R}\right) \tag{12}$$

where R is the radial capillary penetration.

Rearranging and integrating Equation (12) yields Equation (13):

$$R_L = \sqrt{\frac{\varepsilon_V H_P^2 \gamma_{LV}}{3\,\eta_L}\left(\frac{\cos\theta_{PL}}{R_{norm}} - \frac{1}{R_L}\right)}\sqrt{t} \tag{13}$$

where the Laplace pressure is $\Delta P = \frac{\cos\theta_{PL}}{R_{norm}} - \frac{1}{R_L}$, $R_{norm} = R_m\cos\theta_{PL}$. $R_m$ can be calculated by introducing the meniscus radius as $H_P = 2\,R_m\cos\theta_{PL}$.

Adding the meniscus radius to Equation (13), yields Equation (14):

$$R_L = \sqrt{\frac{\varepsilon_V H_P^2 \gamma_{LV}}{3\,\eta_L}\left(\frac{2\cos\theta_{PL}}{H_P} - \frac{1}{R_L}\right)}\sqrt{t} \tag{14}$$

*2.3. Tangential Liquid Transport*

In the third stage, the rate of tangential spreading depends on the pore saturation, liquid surface tension, fluid viscosity, and contact angle. Borhran and Rungta [19] used the Gillespie model [18] to develop a model that predicted the tangential liquid spread using a logarithm plot of the squared radius against the logarithm time (Equation (15)):

$$R_t = R_0\left[1 + k_R\left(\frac{4(u+1)\gamma_{LV}D_{sat}}{uR_0^2\mu L}\right)t\right]^{\frac{1}{2(u+1)}} \tag{15}$$

where $k_R$ is a constant deduced from Gillespie's model—$k_R = k_A\frac{8R_P^2}{27\pi}\left(\frac{\gamma_{LV}}{\eta_L}\right)$ is the rate of radial spreading intrinsic to the material—$R_P$ is the hydrodynamic radii of the capillary liquid radius spread at time t, $R_0$ is the initial liquid radius before spreading, $\gamma_{LV}$ is the liquid–vapor component of surface tension, $\mu$ is the liquid dynamic viscosity, L is the capillary length, $\eta_L$ is the liquid apparent viscosity, u is a material dependent constant, and $D_{sat}$ is a liquid permeability constant.

In essence, the models described above lack a fundamental understanding of nonlinear fluid transport, namely strain recovery, swelling, and solute diffusion, which are the predecessor steps of matrix disintegration. Recent method developments in liquid transport kinetics can correct some of these limitations, as described below.

*2.4. Recent Advancements in Liquid Transport Models*

A recent study by Vaitukaitis et al. [11] mapped the ingress of a liquid into a monodisperse porous structure composed of randomly overlapping spherical particles compacted in a cubical form. The authors used a Lattice–Boltzmann computational fluid dynamics model with a dimensionless Bond number (Equation (16)) and a viscosity ratio (Equation (18)) to characterize the dynamic behavior of fluid penetration.

$$Bo = \frac{\rho g d_0^2}{\gamma} \tag{16}$$

$$M = \frac{\mu}{\mu_g} \tag{17}$$

where μ is the dynamic viscosity of the liquid, $\mu_g$ is the dynamic viscosity of the gas trapped in the porous structure, Bo is the Bond number represented by the ratio of gravitational and capillary effects, and M representing the viscous forces affecting the pore penetration.

The results from Vaitukaitis et al. [11], considering Newtonian fluid properties, showed that the rapid spreading directly correlates with the fluid penetration profile. However, non-Newtonian fluid properties and monodisperse particle sizes can significantly affect the fluid transport in the model proposed by Vaitukaitis et al. [11]. The last section in this review expands the discussion on the effects of fluid viscosity and pore distribution on the disintegration and dissolution kinetics of tablets.

When tablets swell, deviations from Darcy's Law (Equation (8)) can occur in the first stage of liquid penetration. These deviations are a result of pore constriction, which induces changes in the internal pore configuration and, therefore, affects liquid penetration and hydration (Figure 1). Liquid transport through a porous structure has been modified to account for the swelling effects of excipients using a modified version of Darcy's Law with a time-dependent pore structure component [20] as in Equation (18).

$$J_p = -S - \frac{\partial \varepsilon}{\partial t} \tag{18}$$

where $J_p$ is the liquid penetration volumetric flux, S is a sink term corresponding to the rate of absorption, and $\frac{\partial \varepsilon}{\partial t}$ is the rate at which porosity decreases with the swelling of tablets.

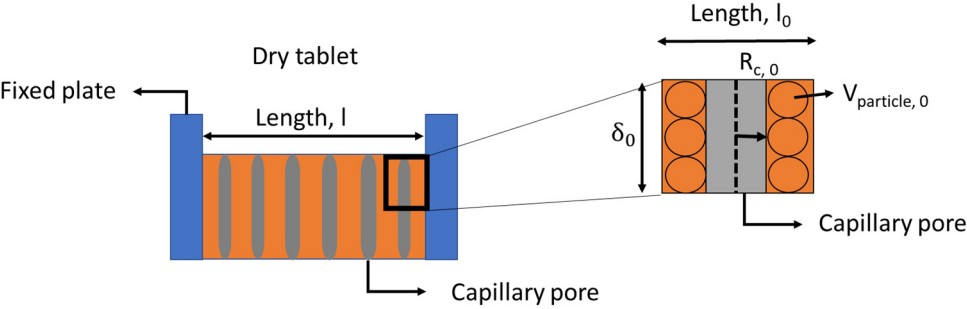

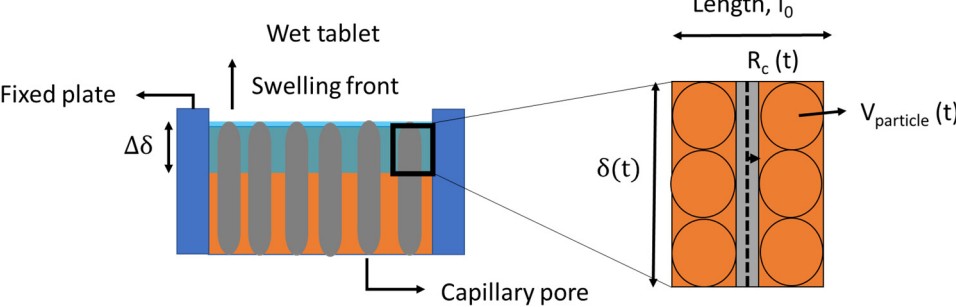

**Figure 1.** Schematics of the effect of matrix swelling on the internal pore structure considering restrained axial swelling. Adapted from: Markl et al. [20]—Open access.

When a tablet swells, it increases in height, which shrinks the capillary radius, $R_C$ (Figure 1). Markl et al. [20] proposed that the fractional increase in volume of the compact is proportional to the fractional increase in volume of a single wetted particle within the compact, $R_C = R_{C,0} - (R_p - R_{p,0})$ (refer to Figure 1). $R_p$ and $R_{p,0}$ are the particle radius after and before wetting, respectively. Equation (20) displays the final correlation between the wetted capillary radius as a function of initial capillary radius and particle sizes as a

function of the initial tablet thickness. The derivation of this equation from Equation (19) is provided in Markl et al. [20].

$$R_C = R_{c,0} - \frac{D_S}{2}\left[\left(\frac{\delta_0 + b \cdot t}{\delta_0}\right)^{\frac{1}{3}} - 1\right]$$ (19)

where $\delta_0$ is the initial tablet thickness prior to axial swelling and b is the swelling rate.

Markl et al. [20] proposed two phases of swelling: (i) the hydration phase accompanied by the release of stored elastic potential in the axial direction (represented by the above Equations (18) and (19)), and (ii) a slower swelling phase that follows the hydration step. The second mode of swelling can be modeled using a modified version of the Schott model [21] that correlates the variation in tablet thickness to the initial rate of swelling (Equation (21)):

$$\Delta\delta = \frac{t}{\alpha + \frac{1}{\delta_\infty - \delta_0}}$$ (20)

where $\delta_0$ represents the thickness of the tablet before swelling, $\delta_\infty$ is the thickness of the tablet after swelling, and $\alpha$ correlates with the initial rate of swelling by Equation (21):

$$\lim_{t \to 0}\frac{d}{dt}(\Delta\delta) = \frac{1}{\alpha}$$ (21)

The term b in Equation (20) can be replaced with $1/\alpha$. In this way, Markl et al. [20] showed the relation between the Schott model and the liquid penetration by the modified Darcy's Law (Equation (21)).

More recently, Martins [22] developed a model to describe the swelling and diffusion concentration dependency due to solute diffusion. This model considered binary mixtures of pharmaceutical excipients and a particle mechanical approach to resolve the system's static mechanical equilibrium under compressive forces. In their work, the constitutive mass transfer of the mobile liquid phase through the porous structure was modeled based on Equation (22):

$$J_{mb} = -4a_{mn}\left(\frac{1}{D_{wm}} + \frac{\chi_{m,n}}{D_{wn}}\right)^{-1}\left(\chi_{m,n}\,\rho_n^w|_\infty - \rho_m^w|_\infty\right) = -\pi a_{mb}^2 H_{mb}^W\left(\rho_w^m|_b - \rho_m^w\right)$$ (22)

where $J_{mb}$ is the flux, $a_{mn}$ is the contact area, w is a subscript for the mobile phase (fluid), b represents the free boundary concentration at the surface $\rho_w^m|_b$ , $\rho_m^w$ is the transport of the solvent inside the particle, D is the diffusivity constant for the mobile phase with respect to particles m and n, and $\chi_{m,n}$ is the partition coefficient, which is a conversion factor of the concentration transfer from particle n to particle m. $H_{mb}^W$ is the mass transfer coefficient, and m and n represent neighboring particles within the porous compact. The overall mass transfer coefficient is described in Equation (23):

$$\overline{H}_{mb}^w = \frac{4}{\pi}\frac{1}{a_{mb}}\left(\frac{1}{D_{wm}} + \frac{\chi_{m,n}}{D_{wn}}\right)^{-1}$$ (23)

The sum of the compressive forces acting on particle m considered the surrounding of particle m by neighboring particle $n \in N_m$ (Equation (24)):

$$\sum_{n \in N_m} P(a_{mn}, \Delta_{mn}, R_m, R_n)\frac{x_m - x_n}{||x_m - x_n||} = 0$$ (24)

where $x_m$ and $x_n$ are the positions of neighboring particles m and n, respectively. The force P is a function of the contact area, $a_{mn}$, particle's relative position, $\Delta_{mn}$, and R represents the radii of particles m and n.

The solute diffusion neglecting swelling effects can be modeled based on transient non-linear particle-to-particle solvent diffusion (25):

$$\left(1 - \frac{\rho_m^w}{\rho^w}\right)^{-1} \frac{M_m^{(s)}}{\rho_m^{(s)}} \frac{\partial \rho_m^w}{\partial t} - \sum_{n \in Nm} \pi a_{mn}^2 \overline{H}_{mn}^W \left(\chi_{m,n} \rho_n^w - \rho_m^w\right) + \sum_{n \in Nb} \pi a_{mb}^2 H_{mb}^W \rho_m^w = \sum_{n \in Nb} \pi a_{mb}^2 H_{mb}^W \rho_w^m|_b \quad (25)$$

This system in Equation (26) can be then converted in matrix form as Equation (26):

$$V(\rho^w, t)\dot{\rho}^w(t) + D\rho^w(t) = J^b(t) \quad (26)$$

where $V(\rho^w, t)$ is the diagonal matrix containing the volume of particles at a given time t, D is the global diffusivity matrix, $J^b(t)$ is the matrix that contains known fluxes, $\rho^w(t)$ represents the matrixes of mass concentration of particles, and $\dot{\rho}^w(t)$ is $\rho^w(t)$ time derivative.

Martins' model [22] considered non-linearities due to liquid transport (i.e., three-dimensional swelling or diffusion concentration dependency) as a first-order differential equation. This system follows an expression analogous to the system in Equation (25) as in Equation (27) with a transient analysis of diffusivity and fluxes:

$$V(\rho^w, t)\dot{\rho}^w(t) + D(\rho^w, t)\rho^w(t) = J^b(\rho^w, t) \quad (27)$$

where $V(\rho^w, t)$ is the diagonal matrix containing the volume of particles at a given time t, $D(\rho^w, t)$ is the global diffusivity matrix, $J^b(\rho^w, t)$ is the matrix that contains known fluxes, $\rho^w(t)$ represents the matrixes of mass concentration of particles, and $\dot{\rho}^w(t)$ is $\rho^w(t)$ time derivative.

A generalized trapezoidal family method was used to time integrate Equation (28). The details of this derivation are provided in Martins' work [22]. Additionally, this model defined a pore-network model to account for the connectivity and geometrical complexity of the pore space using a grain-based model [12]. Contrary to the model presented in Markl et al. [20], the Martins model can map omni-direction swelling of binary component tablets. Furthermore, another advantage of this model compared to Markl et al. [20] is the consideration of contact mechanics effects (i.e., elastic or elastoplastic particles within the tablet structure). Notwithstanding, the solvent diffusion modeling considered Newtonian fluid properties. However, relative to the other models presented in this review, Martins' model can accurately predict the forces generated at the imminence of swelling in binary porous structures such as tablets.

In summary, factors affecting the fluid flow through pore spaces depend on fluid and solid properties such as viscosity, density, the fluid–solid contact angle, surface tension, pore size, shape, porosity distributions, and overall tablet mechanical properties. The fluid transport mechanism in porous media is driven by the capillary pressure and dynamic changes in pore space. The dynamic changes in pore space are specifically applied when considering swelling or diffusion concentration dependency.

## 3. Characterizing Release Kinetics of Tablets

### 3.1. Mechanisms of Disintegration

Disintegration characterizes the breakdown of inter particulate bonds formed during tablet compaction. At high compressive loadings, mechanical interlocking occurs, whereby the bonding mechanism and strength are directly dependent on particle shape, size, and surface roughness of the raw materials [23–28]. Elastic and plastic deformation, particle fragmentation, and the nature of inter particulate bonds can have a direct influence on the tablet's strength and, consequently, the disintegration behavior [5]. The plasticity, viscosity, solubility, dissolution rate, and swelling ability of excipients may also affect the fluid penetration and, therefore, the disintegration kinetics [23].

There can be three major forms of disintegration mechanisms (Figure 2): (i) volume enlargement due to strain recovery, (ii) matrix swelling, or (iii) dissolution of excipients from the pore walls [4].

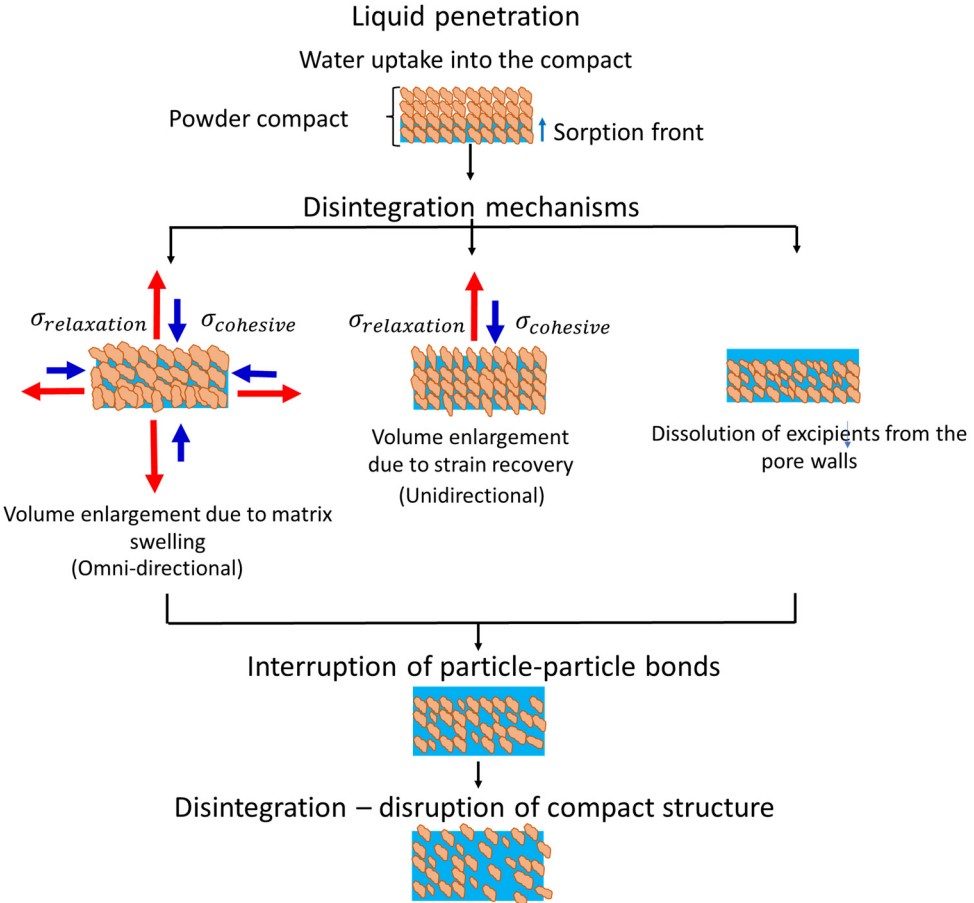

**Figure 2.** Schematics of the disintegration mechanisms. The blue arrowhead ($\sigma_{cohesive}$) represents the conservative stresses withholding the tablet structure, while the red arrowhead ($\sigma_{relaxation}$) represents the dissipative/relaxation stresses triggered by disintegration onset. Adapted from: Markl and Zeitler [3]—Open access.

Strain recovery is a critical parameter as it represents the onset of the disintegration process [26,27]. The critical influence of strain recovery in the disintegration process is because the axial stresses are typically at a maximum threshold compared with the radial stresses [4–6,20,26]. Strain recovery occurs due to the softening of the interlocking between polymer chains or spontaneous crystallization during compaction [7]. The stored energy is then released immediately after compaction in the form of heat or after submersion in fluid media. With hydration, the polymers within the tablet matrix can regain sufficient mobility to recover their original shape.

The limiting rate process of disintegration is characterized by liquid penetration, and, in most cases, this step represents the onset of swelling. Matrix swelling generates the omni-directional enlargement of particles and pressure build-up due to the presence of disintegrants in the formulation. This disintegration mechanism occurs specially in tablets manufactured under high compaction pressure, whereby the sole effect of liquid penetration during the hydration step is not enough to cause bond disruption [4].

The localized dissolution of excipients increases tablet porosity. The increased tablet porosity can induce rapid disintegration onset. In general, the disintegration mechanism iii occurs in the presence of high solubility excipients. In addition to the physicochemical attributes of the excipients in the formulation, liquid penetration is also affected by process conditions. In general, low compaction pressures during tableting produce porous structures that might increase liquid transport and facilitate disintegration.

Hydration, strain recovery, and swelling contribute to the transition of polymers from rigid to soft and rubbery states. During the transition from a rigid to rubbery state, the swelling kinetics change from first- to second-order [20,28]. The presence of strong dipole and high mobility water molecules can break macromolecular hydrogen bonds to further plasticize the amorphous components of the polymers. This plasticizing effect induces higher mobility of polymeric bounds and progression of disintegration. In some cases, however, dense crystalline polymeric structures can cease the accessibility of water molecules and then maintain the cohesive forces between chain segments [29]. In addition to the crystalline–amorphous content, differences in the composition and molecular weight of swelling polymers can also impact the disintegration and dissolution behavior. Kiortsis et al. [30] showed that rapid swelling can either favor or delay disintegration kinetics, depending on the interplay of formulation and process design. The interplay between formulation and process design is discussed in the last section of this review.

Table 1 depicts various methods that are being used to quantify compact disintegration kinetics.

**Table 1.** Methods used to quantify compact disintegration mechanisms.

| Methods | Description | References |
|---|---|---|
| Measuring water uptake Swelling force measurement | Water uptake can be measured based on changes in weight gained by the tablet using a microbalance. The swelling force is measured by fixing the tablet in a cage attached with a load cell, which records the force displacement over time. | Dees (1980) as cited in Markl and Zeitler [4,31] |
| Terahertz pulsed imaging (TPI) | TPI measures the difference in the refractive index of dry material and liquid, resulting in the reflection of the terahertz pulse at the solid–liquid interface. The change in refractive index as a function of time records the liquid penetration through the porous matrix. The pore size, shape, and compact density can also be assessed using this technique. The swelling rate can be determined from the shift in reflectance. | [8] |
| Erodibility | Erosion of porous tablets can be evaluated by weighing the tablets upon drying and removing excess liquid. | [32] |
| X-ray microtomography (XRµT) | Based on the principle of voxelization, 2-D X-ray images are reconstructed and converted into 3-D images. Pore volume, size distributions, degree of anisotropy, and the distinction between open and closed pores can also be computed with this technique. The porosity distributions as a function of dissolution time can be measured offline by freeze-drying samples before the analysis. The pore size and density can be plugged into the capillary transport models to map the liquid distribution. The swelling ratio can also be computed using the changes in volume, measured using XRµT, and the change in weight. | [33,34] |
| Nuclear magnetic resonance (NMR) | The NMR technique can be used to map the movement of water in a solid matrix and calculate the diffusion coefficient by studying the decrease in the solid's NMR spectra signal and the increase in the liquid's NMR spectra. NMR imaging instruments can be used to map the swelling kinetics and diffusion behavior of water in dry compacted tablets. | [35,36] |
| Magnetic resonance imaging (MRI) | In this technique, 2D or 3D images are obtained by placing a sample into a magnet and varying the field strength and the frequency of pulses over time and space. The result of the analysis represents a distribution of the sample protons at different phases and frequencies. The free induction decay can be analyzed using multidimensional Fourier transformation to produce spatial slices of the sample. The images can present information, for example, about the water distribution patterns in solid samples and the changes in granular microstructures as a function of the water ingress. | [36,37] |

**Table 1.** *Cont.*

| Methods | Description | References |
|---|---|---|
| Near Infrared Spectroscopy (NIR) | NIR is used as an in-line process analytical tool to estimate the number of compounds in a formulation, the level of strain recovery, the amount and particle size of formulation constituents, the ratio between polymorphic forms, and compression force in direct compression experiments. This is a robust technique that provides information for the prediction of the disintegration and dissolution kinetics. | [2,38–43] |
| Texture analyzer | Force displacement profiles can be directly measured using the penetration of a probe within a compact. The force profile is directly proportional to the strength of the tablets. A disadvantage of this technique over NRI, microCT, and NMR-MRI is the lack of detail on the porosity distribution and swelling kinetics. | [40] |
| Visible and UV-dissolution imaging | This technique is widely used in intrinsic dissolution tests and transport studies and has recently been used to investigate drug liberation mechanisms and excipient properties, such as swelling kinetics. The reduced imaging area restricts its use to small-size tablets. | [41–43] |

### 3.2. Mechanisms of Dissolution

Drug release patterns can either be (i) immediate, (ii) extended, or (iii) delayed. Examples of an immediate-release formulation with a rapid drug release are provided in Table 2, such as orally disintegrating tablets, effervescent tablets, and some types of buccal or sublingual tablets. The internal pore structure is a critical factor in the design of immediate-release formulas because of its direct relationship with disintegration kinetics. The fast-disintegrating characteristic of these immediate-release formulas maintains the therapeutical dosage levels for a short period in the plasma. These immediate-release tablets require maintenance of mechanical strength until administration [44]. Therefore, tablet microstructure design is also critical to ensure product stability, while providing an ideal immediate-release function.

**Table 2.** Examples of tablet formulations with different release mechanisms.

| Type of Oral Dosage Form | Description |
|---|---|
| Orally disintegrating tablets | Offer rapid disintegration within the oral cavity in the presence of salivary fluid. They are generally produced using freeze-drying or loose compression and recently using 3-D printing technology to improve dosage content. These tablets can be used as immediate-release agents. |
| Lozenges | Require slow dissolution in the mouth, typically dedicated to the relief of cough, sore throat problems, mild anesthesia, or antiseptics. These tablets can be used as extended-release agents. |
| Layered tablets | Multi-layered tablets containing multiple layers on top of each other and press-coated tablets whereby a core tablet is enclosed with a shell tablet. These tablets can be used in extended- and delayed-release applications. |
| Effervescent tablets | Intended to release carbon dioxide upon immersion in water, which prompts its disintegration. The chemical reaction to produce $CO_2$ is a result of the interaction between carbonate or bicarbonate salt and a weak acid in the presence of water. Upon ingestion, the gastric pH levels enable rapid drug absorption from the upper small intestine. These immediate-release formulas are often manufactured using direct compression or with granulation followed by compaction. |
| Buccal or sublingual | The mechanism of drug absorption is by positioning the tablet into the internal mucosa of the cheek (buccal) or under the tongue. The drug is absorbed through systemic circulation. The surface area of these tablets is usually small, ensuring good adhesion properties. Gums or bioadhesives can be added to improve adherence. These tablets also possess higher porosity for fast disintegration and absorption of the drug and are, therefore, classified as immediate-release products. Polymers with a melting temperature close to body temperature can also be added to the formula to trigger rapid glassy–rubbery transition and, hence, fasten drug dispersion. |

Extended dosage forms slowly release the drug for prolonged periods, minimizing dosing requirements, such as lozenges and layered tablets (Table 2) [44–48]. The four operational modes of release extension include dissolution-controlled, diffusion-controlled, osmotically, and ion-exchange controlled [44]. The delayed-release formulations are intended to target a specific region of the gastrointestinal tract for localized action through functionalized induction, such as release based on changes in pH. The performance of these formulas is more dependent on the properties of the outer/coating layer and/or environment-induced properties to extend or control the release kinetics rather than the tablet's internal pore array [44]. Table 2 includes examples of tablets used in clinical practices.

Dissolution in immediate-release formulas occurs after the disintegration process. In general, dissolution is inversely correlated with disintegration time in immediate-release formulas [49]. A schematic of active ingredient release in immediate-release tablets is provided in Figure 3. Dissolution is a process in which a mixture of two phases (the solute and the solvent) produces a new homogeneous phase: the solution [50]. The degree of homogeneity of the solution depends majorly on the chemical and physical characteristics of the components and the environmental conditions, such as pH, temperature, and pressure. During dissolution, the surface area of the solute exposed to the liquid solvent diffuses to the outer media; this interaction between solute and solvent characterizes the process of solvation [51]. This process continues until solvation and precipitation reach an equilibrium state, at which point the solvent can no longer dissolve the remainder of solute particles. Dissolution models often treat dissolution as an inflexible process, disregarding the surface area–volume change due to the generation of undissolved particles. Therefore, model optimizations able to account for this dynamic effect are fundamental to controlling dissolution onset [52]. For instance, dissolution models that predict particle fragmentation can modulate the disintegration process and ensure an effective immediate-release mechanism. Recently, Seager et al. [52] developed a model for characterizing the dissolution of a solid considering the dependency of diffusion by the interplay of surface area and physical fragmentation as in Equation (28):

$$\frac{\partial N(V,t)}{\partial t} = -\frac{\partial}{\partial V}(R(V,t)N(V,t)) + \int_{V}^{V_0} g(V_i)f(V_i \rightarrow V) \cdot N(V_i,t)dV_i - g(V)N(V,t) \quad (28)$$

where $R(V)$ is the diffusive volume removal function based on the Nernst–Brunner model, $g(V)$ is the fragmentation rate and $f(V_i \rightarrow V)$ is the transition function that encodes the distribution of particles of volume $V_i$ generated by the series fragmentation of the original particle volume, $V$.

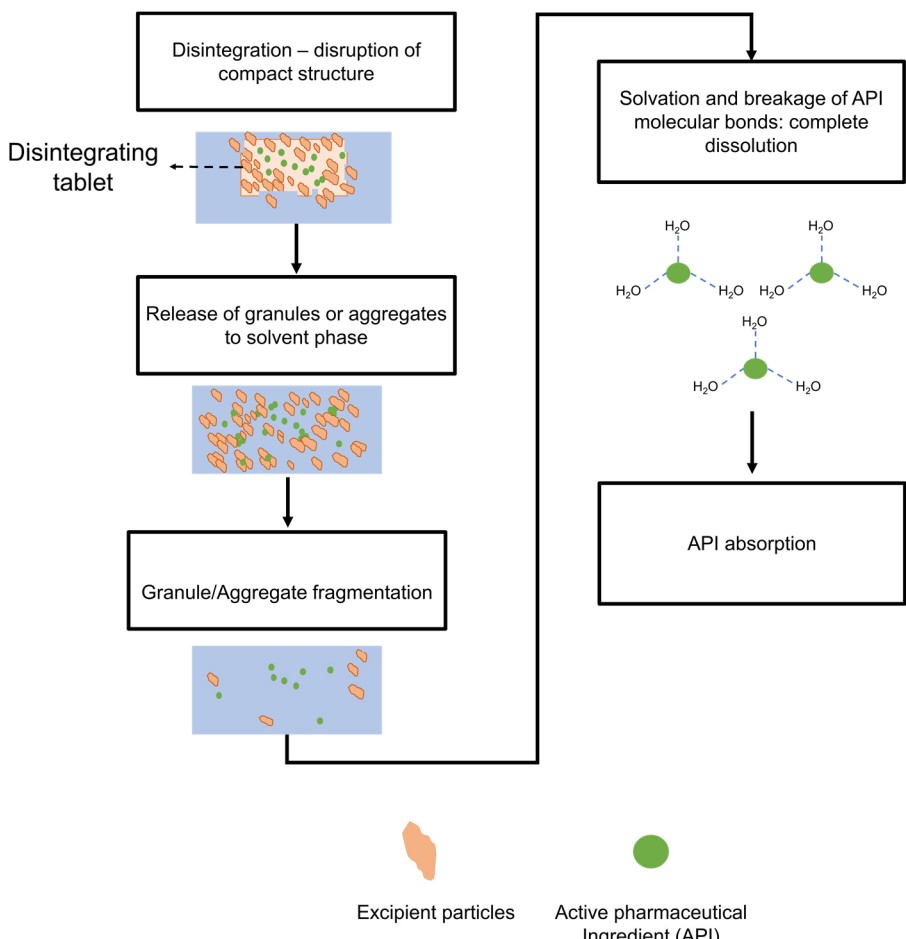

**Figure 3.** Schematic of active pharmaceutical ingredient (API) release in immediate-release tablets.

## 4. Dependence of Liquid Transport Kinetics on Tablet Microstructure Design

The tablet structure is affected by primary particle size, shape, plasticity, and bond strength between different excipients and APIs. The process parameters, including compressive force and dwell time (referred to as the time of applied loading), also control the compaction process and the resulting porosity distribution of tablets. Disintegration and dissolution are also influenced by physicochemical attributes such as the diffusivity, solubility, and swelling ability of excipients. With an understanding of the microstructure role in the tablet properties, the optimization of formulation and process parameters can be more easily achievable. The following sub-sections explain the effect of material properties and processing conditions on the pore structure and liquid transport through the pores.

### 4.1. Effect of Processing on Tablet Microstructure

4.1.1. Intermediate Granule Properties Affected by Dry or Wet Granulation

Wet or dry granulation may precede the dry compaction step to improve densification and particle contact during tableting [30]. The characteristics of intermediate granules produced with granulation can influence tabletability [53–56] and, consequently, the quality attributes of immediate-release tablets, such as the strength and disintegration kinetics. For instance, low porosity granules yield weaker tablets due to work hardening. The work hardening renders the material resistant to additional plastic flow. The drying process type and drying rates post-wet granulation can affect the granule's shape and internal pore structure. Therefore, intermediate granule properties should be carefully selected to ensure optimal compressibility and compaction during tableting, while maintaining acceptable disintegration and dissolution onset. Tables 3–5 summarize the effect of granulation methods on the intermediate granule microstructure design.

**Table 3.** General process design targeting granule properties in dry granulation using the roll compaction technique.

| Formulation/Process Design | Design Characteristics |
|---|---|
| Roll hydraulic pressure (kPa) | High roll pressures achieved by adjusting the gap between the rollers can increase the interparticulate bonding and accelerate material densification, consequently generating ribbons with superior relative densities (low porosity). By evaluating the interaction between the input parameters using the Johanson's rolling theory coupled with mass balance in the nip region, Hsu et al. [57] stated that the roll pressure exhibits the most significant effect on the ribbon porosity. Roll pressure can also influence granule properties, including particle size and porosity |
| Roll gap | Roll gap defines the thickness of ribbons and can influence the porosity of ribbons, but this parameter is highly correlated with roll hydraulic pressure, roll speed, feed, and tamp auger speeds. Therefore, caution should be exerted when manipulating this variable to ensure proper consideration of the effect of hydraulic pressure and roll speed on the roll gap. |
| Roll speed (rpm) | A reduction in roll speed can produce ribbons with low porosity, depending on the formulation, because of a rise in contact time of the densifying material within the nip region [57,58]. |
| Feed speed (rpm) | The design of the control system should include only one of the speeds as the manipulated variable to maintain a constant ratio between feed speed and roll speed. If this ratio is not constant, then slippage and burning of the powder material between the rollers can lead to product degradation. |
| Granulator sieve types | Granulator screens are selected to determine granule size. Milling occurs clockwise and counterclockwise in oscillating screens. Meanwhile, milling occurs only in one direction in rotating screens. In general, both screen types generate comparable particle size distributions. |
| Granulator speed (rpm) | An increase in granulator speed can reduce granule particle size. The effect of granulation speed on granule porosity depends on the ribbon relative density. |
| Granulator type (−) | There are two main modes of granulators: abrasive fracture and impact fracture. Both methods produce granules with similar particle size distributions and granule strength. |

**Table 4.** General process design targeting granule growth and consolidation regimes for high shear wet granulation (high shear mixing and twin-screw granulation—TSG).

| Formulation/Process Design | Design Characteristics |
|---|---|
| Feed powder porosity (−) | Porous powders ensure more consolidation in both high and low shear techniques. While less porous powders produce less consolidation and, therefore, more porous granules. For instance, high porosity of primary particles (e.g., MCC) result in enhanced liquid absorption. The stored liquid content is squeezed out of the powder mass upon shearing, enabling granule consolidation [59]. |
| Liquid to solid (L/S ratio) (−) | High values increase deformability, which varies with formulation (e.g., if the binder is more hydrophilic, less solvent is necessary to promote the binder distribution in the high shear granulation processes). Low values decrease deformability and, therefore, produce granules with higher porosity [60]. |
| Binder viscosity (Pa s) | The viscosity of a binder dictates the strength and size distributions of the resulting granules. Studies unveiled thresholds of viscosity at which granule growth was favored, and an opposite effect was observed after surpassing a critical value [61]. Granulation at low viscosities is controlled by layering growth, while granule coalescence is predominant at high viscosities [62]. |
| Surface tension (N/m) | Ivenson et al. [63] found that a decrease in binder surface tension increases the consolidation rate due to a reduction in capillary suction resisting particle dilation. They also showed that a decrease in particle size or an increase in binder viscosity can lower the consolidation rate. |
| Binder wettability (°) | Defines better interaction between the binder and primary particles; better liquid distribution endows more consolidation in both high and low shear wet granulation. |
| Velocity profile (rpm, m/s) | In high shear mixing processes, a decrease in speed generates less densification and liquid distribution. Meanwhile, an increase in speed can generate more granule consolidation. Seeded granules are also formed more rapidly with a high Stokes deformation number [63]. |
| Powder feed rate (kg/s)/Residence time (s) | In high shear mixing processes, this parameter has varying outputs in the literature. |
| Screw geometry in TSG | Higher granule consolidation in twin screw granulators using kneading elements. However, kneading elements may also cause more granule breakage in size ranges between 2 and 3 mm [64]. |

**Table 5.** General process design targeting granule growth and consolidation regimes for low-shear wet granulation processes (focus on fluid bed granulation—FBG).

| Formulation/Process Design | Design Characteristics |
|---|---|
| Distance from the nozzle in FBG (m) | In FBG, particles closer to the nozzle position produce more agglomerates [65]. |
| Bed temperature in FBG (°C) | Higher bed temperatures disrupt the liquid bridge as the binder solidification rate may reduce significantly, especially above binder melting temperature. However, adjustment of the air inlet temperature and binder flow rate can improve granule consolidation because of the effect of these two parameters on the drying potential [65,66]. |
| Binder flow rate in FBG (kg/s) | Aggregation is improved with an increase in binder spray rate [66]. |
| Binder droplet size in FBG (m) | Larger droplet size improves wettability and, therefore, the probability of particle coalescence or aggregation [66]. Better liquid distribution improves granule consolidation and growth. |
| Atomizing pressure in FBG (bar) | Lower atomizing pressure reduces consolidation rate; droplet size is negatively correlated with atomizer pressure [66]. |
| Riser dimension (m) | In a Wurster fluid bed with nozzle positioned at the bottom of the bed, smaller riser diameter implies narrower particle size distributions. This is because of stable flow across the bed, which reduces the gap between the residence time of smaller and larger particles [65]. |
| Drying potential | Slow drying rates result in more porous granules using fluid bed spray granulation (coating deposited onto seed particle's surfaces). A balance between air inlet temperature, velocity profile, air humidity, and binder flow rate need to be considered to obtain suitable granule consolidation but still yield granules with controlled porosity [65,66]. |
| Powder feed rate (kg/s)/Residence time (s) | In fluid bed granulation, reduced residence time seems to decrease volume solid fraction [65,66]. |
| Velocity profile (m/s) | Fluid bed granulation studies show that high velocity profiles can cause particle breakage [66]. |

### 4.1.2. Uniaxial Compression/Tableting

During tableting, a traditional uniaxial compression method for tablet production, powders or granules are compressed at determined conditions until forming a continuum compacted structure. At the onset of compression, depending on surface properties and frictional effects, particles change their packing state without excessive deformation. In addition, mechanical interlocking and van der Waals forces are dominant in this initial stage of consolidation. Elastic and plastic deformation occur with the progression of applied load, resulting in material densification and particle fracture [53,54]. This stage of consolidation, denominated compaction, defines the inter particulate bonding and tablet pore arrangement. Formulation parameters, such as particle size distribution, shape, plasticity, and bonding probability between different excipients and APIs are among the particle properties that influence the attrition, compactibility, densification, and final bed void space of the compacted structures.

In addition to upstream granulation processes and tableting process described above, stress relaxation also affects the void space in a compact. Stress relaxation, characteristic of

the viscoelastic properties of the excipients, is related to the tablet's plastic recovery after unloading. The tablet's void space, which is a function of stress relaxation, constitutes the random architecture of the pore structure. The tablet's void space is affected by a combination of process parameters, such as pressure, compaction speed, dwell time, and formulation properties. For instance, tablets produced using the same compressive loading and under the same speed, dwell time, and relaxation time may present the same bulk porosity. Yet, these tablets can have distinct fluid flow and internal pressure build-up. These distinctions in fluid flow are associated with variations in the tablet void space affected by the stress relaxation behavior [30].

### 4.2. Effect of Formulation Design on Disintegration and Dissolution Kinetics

4.2.1. Influence of Porosity

The complex characteristics of pore systems invalidate the use of bulk porosity as the sole indicator of the compaction process. Therefore, the degree of compression, affected by attrition, deformation, and densification, is most suitable to characterize the compaction process [54]. The relationship between tablet pore structure and raw material characteristics (granule/powder porosity and mechanical properties) is well documented in the relevant literature [54,55]. Materials with low yield pressure improve bed mobility during compaction, which facilitate densification [67–70]. However, the incompatibility of interparticle bonding between excipients and API is majorly responsible for weakening compact strength [68]. As discussed in previous sections, compact strength might play a role in the fluid penetration and disintegration kinetics.

The use of polydisperse size ranges of raw materials can increase a tablet's bond strength and, therefore, delay the disintegration time [69]. The higher bond strength is due to the interlocking of fine and coarse particles. Additionally, intermediate granulated powders with high porosity generally exhibits better fragmentation and attrition properties [71]. Perez-Gandarillas et al. [23] observed that different particle size distributions of microcrystalline cellulose grade PH101 used as feed material presented distinct deformation during dry compaction. This signified that additional pressure was required on the fine feed material to produce the same plastic deformation as in the granular feed forms.

In general, sharp-edged particles may reach mechanical stability quicker than spherical particles. He and Guo [25] and Yohannes et al. [6] noted that the presence of sharp-edged particles led to distinct particle rearrangement, which improved attrition/fragmentation and consolidation behavior; therefore, generating tablets with less bed void space and higher mechanical strength.

Mechanical properties of excipients and active components also have a significant influence on the tablet microstructure. For example, brittle materials with higher attrition/fragmentation behavior tend to produce tablets with smaller pore sizes, while plastic materials produce less porous tablets because of their higher deformability and higher binding ability. Skelbaek-Pedersen [7] found that tablets manufactured with coarser size fractions of microcrystalline cellulose (MCC) exhibited lower liquid penetration compared with tablets using fine MCC particles. Meanwhile, di-calcium phosphate tablets did not exhibit significant changes in liquid transport as the fragmentation ability was similar for all the size fractions considered in their study.

In general, less porous tablet structures lead to a delay in the disintegration rate processes. As previously discussed, liquid penetration is the rate-limiting process that initiates strain recovery, swelling, localized dissolution, and, finally, tablet disintegration [20]. Nonetheless, additional factors associated with material physical and chemical properties, such as viscosity and hydrophilicity of excipients and APIs, influence fluid flow through the pore space. For instance, liquid absorption increased in tablets with HPMC 2208 because of small particle size and low viscosity [72]. This example highlights the influence of composition on liquid transport kinetics even when the tablets present similar bulk porosity before submersion in fluid media.

### 4.2.2. Influence of Excipient Properties

Disintegration and dissolution onset depend on excipient properties (e.g., mechanical properties, hydrophilicity, solubility, viscosity) and overall tablet strength. The typical excipients used in immediate-release formulas are disintegrants, fillers, binders, and lubricants. Chemical substitutions in excipient side chains can explain the interplay between excipient properties and tablet properties. These chemical substitutions may increase excipient molecular weight. A wide range of polymer properties is dependent on molecular weight, such as viscosity [73,74], glass transition temperature, and mechanical properties. A higher molecular weight polymer may delay dissolution as the gain in entropy from the polymer dissolution remains low [73]. The consequences of low entropy are that low-energy intermolecular forces, such as van der Waals forces, can generate strong interchain associations. These associations result in hydrophobicity across the chains, hindering solvation and dissolution [73]. However, depending on the interplay between tablet strength and excipient properties, a higher molecular weight excipient can expedite swelling kinetics and, consequently, accelerate the disintegration rate due to rapid internal pressure build-up. This variability in internal pressure build-up is explained below based on the different wettability, dissolution, and swelling characteristics of excipients. These characteristics are attained with side-chain substitution and crosslinking mechanisms that affect the mechanical and physicochemical properties of polymeric structures used as excipients. The variation in physicochemical properties of excipients directly affect the rate processes involved in disintegration and dissolution kinetics.

Wettability, dissolution, and swelling controlled regimes classify the disintegration mechanism in immediate-release formulas [75]. A schematic representation of the disintegration mechanisms limited by formulation aspects is shown in Figure 4.

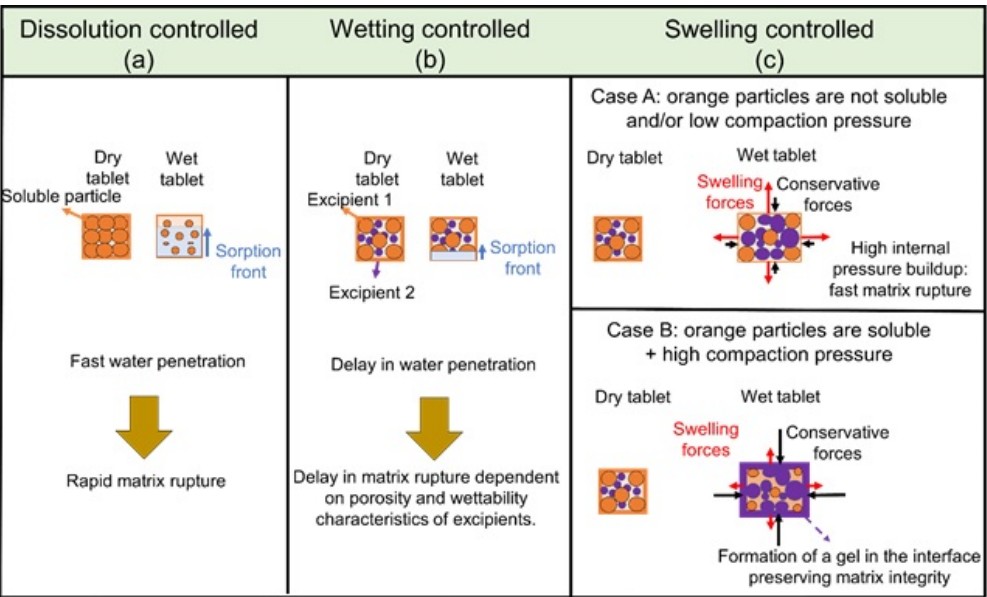

**Figure 4.** Schematics of disintegration mechanisms as affected by formulation characteristics for immediate-release tablets: (**a**) Dissolution controlled disintegration regime whereby rapid dissolving excipient particles contribute to faster liquid sorption (blue arrowhead) and rapid disintegration onset, (**b**) Wetting controlled regime whereby the disintegration onset is limited by excipient wettability and tablet porosity, and (**c**) Swelling controlled regime whereby the disintegration onset is limited by excipient solubility and overall tablet strength. The black arrowheads in the swelling controlled regime (**c**) indicate conservative forces withholding the tablet while the red arrowhead indicates the dissipative forces (swelling force in this case) triggering disintegration onset.

Changes in tablet porosity steadily influence wettability-controlled formulations. The tablet integrity depends on the wetting properties of the excipient and its overall strength

in formulations containing excipients with rigid molecular chains [76]. Meanwhile, hydrophobic lubricants, such as magnesium stearate, can reduce tablet wettability while significantly compromising the tablet breaking force. In dissolution-controlled regimes, the rapid dissolution rates of filler components may trigger additional pore formation in the tablet structure and, consequently, accelerate liquid penetration [77,78].

The concentration and type of disintegrant steadily affect the disintegration time in swelling-controlled systems. In this case, Maclean et al. [75] observed that a swelling force able to generate matrix rupture is dependent on the overall tablet composition rather than the porosity. For instance, high porosity tablets containing microcrystalline cellulose (MCC) and mannitol exhibited slower disintegration kinetics. This trend is because fast liquid penetration yielded rapid dissolution of mannitol, generating more void space in the tablet structure. With additional void space, the swelling of filler particles (mannitol) occurred locally and did not exert pressure against the surrounding MCC particles. Arndt and Kleinebudde [71] also correlated gelling capacity to a delay in disintegration time. The swelling kinetics and gelling properties of excipients have an interesting correlation with the pore structure that can affect the internal forces developed pre-disintegration onset. The prediction of internal forces as a function of formulation and process can improve the design of an immediate-release response.

Different grades and combinations of super disintegrants can also modify disintegration and dissolution patterns. For instance, the high wetting profile of hydrophilic disintegrants can expedite the dissolution of a drug component [1]. The particle size distribution of disintegrants, in addition to the physicochemical properties, generally drive the disintegration and dissolution kinetics. Larger particle sizes of specific types of disintegrants may reduce diffusivity, hence, delaying disintegration and dissolution onsets [1].

The Quality-by-Design (QbD) approach has gained momentum in the design of immediate-release tablets. Generally, this approach consists of mapping material attributes and process parameters based on a risk assessment. This risk assessment can be realized based on preliminary analytical and modeling investigations and previously available studies to define the Quality Target Product Profile (QTPP). The QTPP is critical to understand and define the formulation and process design based on clinical use, route of administration, formulation, delivery mechanism, dose requirements, and packaging. The selection of critical material attributes (CMAs) and critical process parameters (CPPs) affecting formulation can be selected from the risk assessment analysis. The range of CMAs and CPPs can be optimized with a Design of Experiments (DoE) containing a central design. The central design within the DoE is likely to yield the desired critical quality attributes for the immediate-release product, such as mechanical properties, disintegration, dissolution, content uniformity, and impurities. Lee and Kim [79] developed a design space for rabeprazole sodium dry-coated tablets for the treatment of gastroesophageal reflux disease (GERD). Oh et al. [80] developed a QbD study of telmisartan potassium tablets, which are bioequivalent to the commercially available Micardis (telmisartan free base) tablets. Oh et al. [80] found that kneading time in the wet granulation process, binder type, and disintegrant type were critical parameters affecting the CQAs (disintegration, tablet breaking force, friability, drug dissolution, and impurities). Misha and Rohera [81] developed a design space using the QbD approach and statistical formulation design for carbamazepine orally disintegrating tablets. They found that compaction pressure, the concentration of the sublimating agent, and disintegrant concentration significantly affected the CQA, such as tablet crushing strength, tablet porosity, disintegration time, water absorption time, tablet friability, and drug dissolution. Finally, two general diagrams to study disintegration and dissolution kinetics in immediate-release tablets based on formulation and process design are summarized in Figures 5 and 6. Most pharmaceutical development studies are mainly interested in the bulk-level properties, namely, friability and tablet breaking force. However, these studies lack a fundamental investigation of micro-level characteristics affected by the properties of raw or intermediate material and overall tableting conditions. Studies

on the effect of formulation and how it alters tablet microstructure and fluid penetration may provide a mechanistic approach to facilitate the manufacturing process and optimize the release of therapeutic agents. Additionally, the QbD approach can be used as a robust tool to realize the desirable quality attributes complying with the QTPP and overcome the technical and commercial challenges in developing immediate therapeutic formulas.

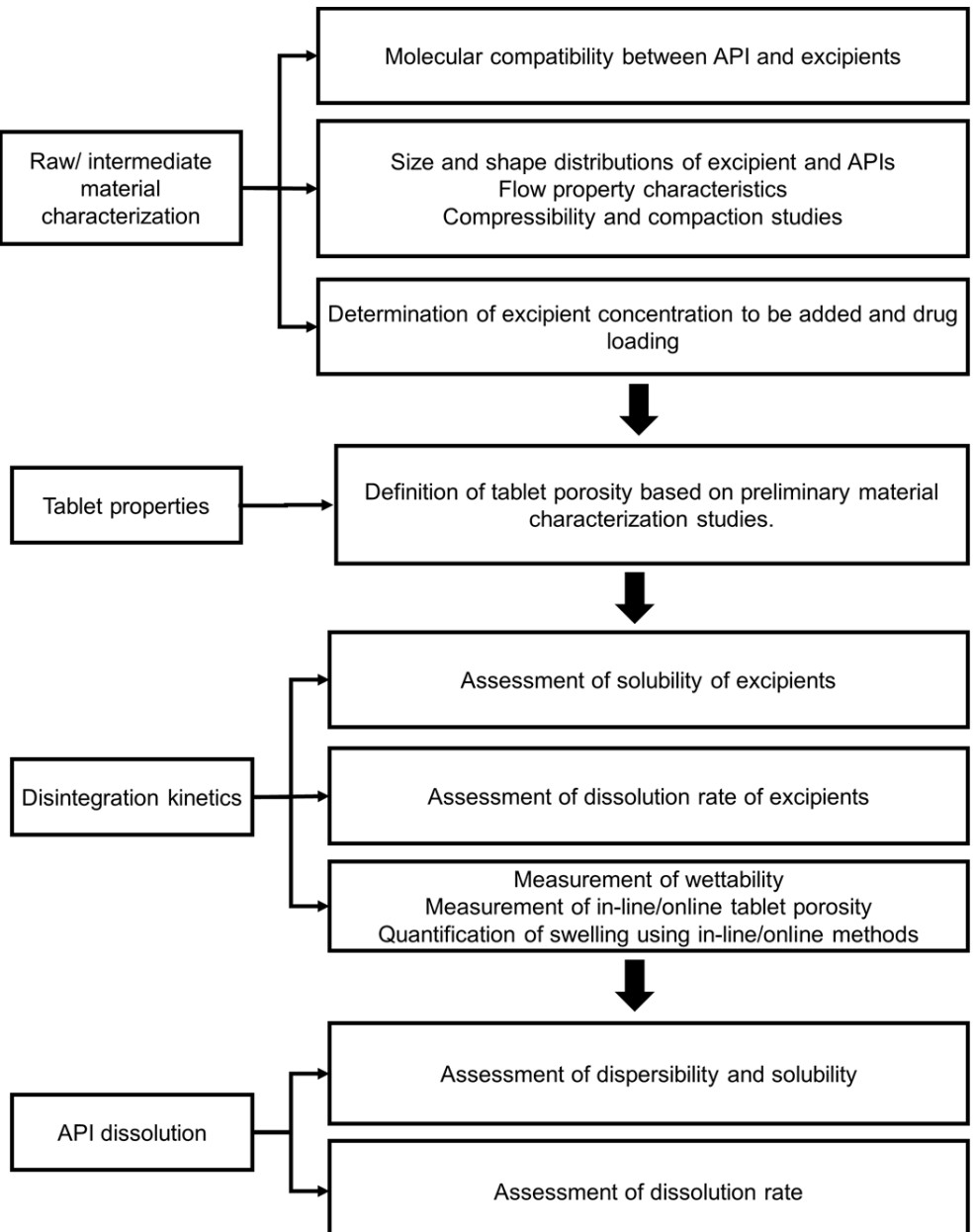

**Figure 5.** Stages for assessment of the interplay between formulation and tablet microstructure on the disintegration until complete dissolution of the API from immediate-release tablets.

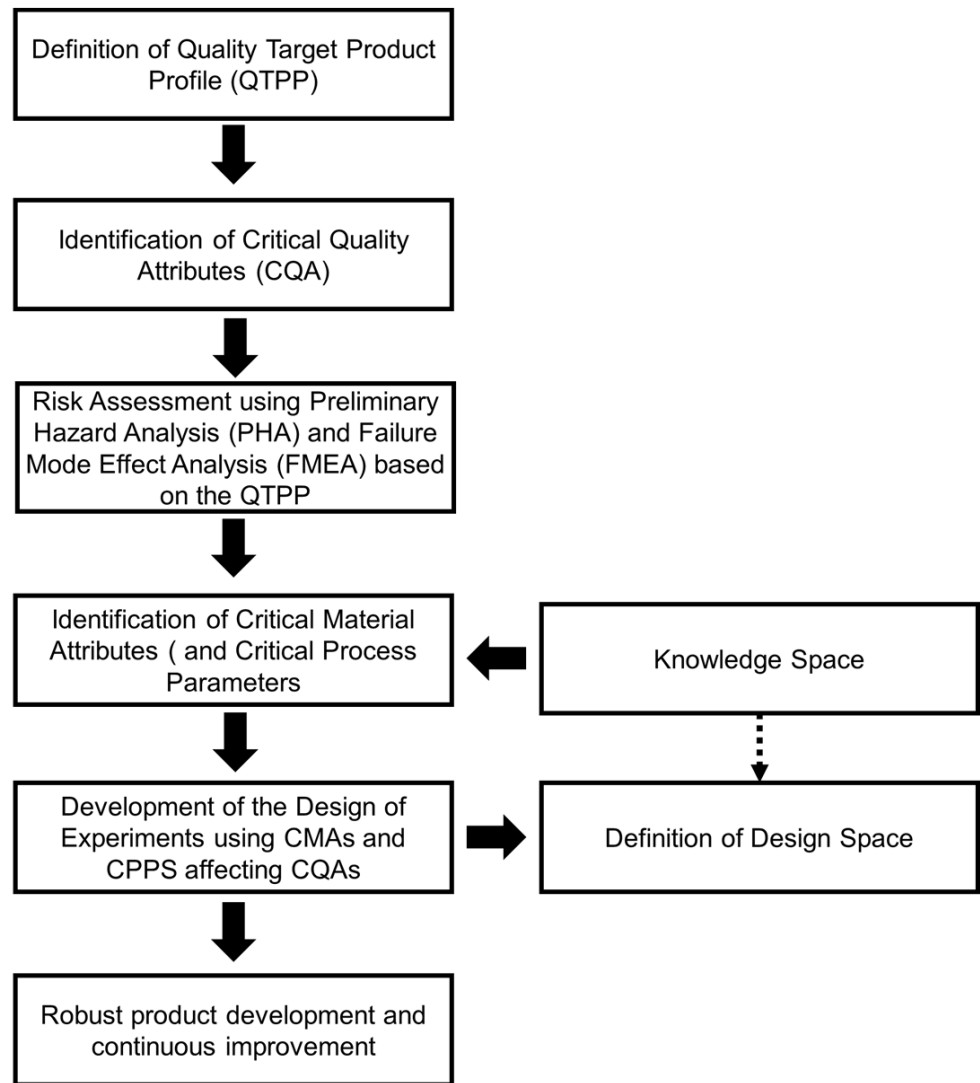

**Figure 6.** Steps in the development of immediate-release formulas based on a Quality-by-Design approach.

## 5. Conclusions

Formulation and process design and their effect on the bulk properties of pharmaceutical tablets have been extensively explored in the relevant literature. However, the role of these parameters on the internal pore structure lacks mechanistic understanding. The topics covered in this review included fluid transport fundamentals and a general mechanism of disintegration and dissolution rate processes to achieve specific release profiles matched by immediate-release formulas. A general discussion on formulation and process attributes affecting the tablet microstructure was also presented. Examples of Quality-by-Design (QbD) studies have also been provided.

Opportunities for optimization and innovation in microstructure design derive from the intricate nature of formulation and process complexities. For instance, there is still a need to establish a design space relating upstream process type (dry, wet granulation, or direct compression), process variables, and formulation attributes to final tablet properties, such as porosity, disintegration, and dissolution onset. The effect of different excipient physicochemical properties, such as swelling, solubility, and interfacial properties, and their effects on tablet pore array before and during disintegration onset, are critical to assessing matrix integrity and tuning the release profile in immediate-release formulas. Moreover, the role of excipients, such as fillers, binders, and lubricants, in boosting dis-

integrant performance is another aspect of formulation-process integration that requires further investigation.

Finally, a primary focus on theoretical, analytical, and modeling analyses of swelling and formulation properties such as wettability, solubility, and dissolution rate is critical to assess tablet disintegration and tune the release profile. This investigation can aid in quantifying internal forces developed at the imminence of tablet disintegration. This review provided insight into the importance of microstructure design in minimizing process inefficiencies and controlling immediate-release tablet properties.

**Author Contributions:** Conceptualization, C.G.J., C.R.W., and K.A.; methodology, C.G.J.; formal analysis, C.G.J.; writing—original draft preparation, C.G.J.; writing—review and editing, C.R.W. and K.A.; supervision, K.A. All authors have read and agreed to the published version of the manuscript.

**Funding:** This research received no external funding.

**Institutional Review Board Statement:** Not applicable.

**Informed Consent Statement:** Not applicable.

**Data Availability Statement:** Not applicable.

**Conflicts of Interest:** The authors declare no conflict of interest.

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
