# Peer review of "The Significance of Tablet Internal Structure on Disintegration and Dissolution of Immediate-Release Formulas: A Review"

_2674-0516, doi:10.3390/powders2010008_

Round 1

Reviewer 1 Report

The manuscript by Jange et al. described the The Significance of Tablet Internal Structure on Disintegration and Dissolution of Immediate Release Formulas: A Review. Authors have written the review paper well and this review paper is suitable for the publication in this journal.

However I did not get any details regarding the direct quantification of tablet microstructure using a Quality by Design (QbD) approach or importance of QbD approach for this review work. Authors need to add this QbD part in the manuscript.

Author Response

Comment

Manuscript response

The manuscript by Jange et al. described the The Significance of Tablet Internal Structure on Disintegration and Dissolution of Immediate Release Formulas: A Review. Authors have written the review paper well and this review paper is suitable for the publication in this journal.

However I did not get any details regarding the direct quantification of tablet microstructure using a Quality by Design (QbD) approach or importance of QbD approach for this review work. Authors need to add this QbD part in the manuscript.

Thank you for this important comment. In agreement with the reviewer’s comment, a phrase in the introduction (lines 68-70, page 2) has been included to mention that the QbD approach is discussed in the final section of this review. In addition, a paragraph discussing QbD approach in immediate release products has been added in section 4.2, sub-section (4.2.2, lines 657-679, pages 21 and 22). An additional figure has been added in this section (Figure 6) detailing the steps followed in the development of immediate release products using a QbD approach.

Reviewer 2 Report

The submitted manuscript presents a review focused on factors influencing tablet disintegration and dissolution, particularly the relationship between tablet microstructure and its disintegration/dissolution, as well as methods used for characterization of these properties. The manuscript provides interesting insight into the literature findings related to this topic. However, the following points should be considered.

There are numerous omissions throughout the manuscript regarding the reference citing. i.e. the numbers enclosed in parentheses do not agree with the numbers of the corresponding references in the list. Please carefully check the reference list and all numbers in parentheses within manuscript text.

Lines 49-50. What is meant by compression strength? Tablet mechanical strength is usually characterized by the breaking force (“hardness” as a term is not preferred, please refer to USP), while compression pressure is important process parameter that can affect tablet mechanical properties (e.g. tablet breaking force, friability).

Line 87. In order to more clearly distinct between liquid properties and characteristics of tablet, please correct this part as follows: liquid properties and tablet's microstructural properties.

Line 414-415, line 421. The meaning of the following phrases is not clear: "high normal load compacted tablets", "low normal loads". Please explain and correct. 

Line 441-448. Immediate release preparations and rapid drug release are not typical only for orally disintegrating tablets, and therefore this paragraph might be misleading and should be corrected. There is a variety of different solid dosage forms showing immediate release of the API. Typical property of orally disintegrating tablets (orodispersible tablets) is rapid disintegration within the oral cavity in the presence of salivary fluid.

Table 3. The parameter L/S ratio should be defined.

Table 4. The abbreviation FBG has not been introduced previously.

Table 4. The influence of binder droplet size in FBG is not clear. Please explain the influence of droplet size on wettability.

The following references are cited in Table 4, which presents the influence of process parameters for wet granulation in fluid bed granulator: Tan et al. Powder Tech. 2004, 143–144, 65–83; Chua et al. Chem. Eng. Sci. 2013, 98, 291-297. However, these studies are related to hot melt granulation that is considerably different process, with different granulation mechanisms involved and different granule morphology and microstructure.

Caption for Table 4 should be corrected considering that only influence of process parameters was discussed not the influence of formulation parameters.

Line 627. Mannitol can be filler not disintegrant. Please correct this.

Author Response

Comment

Manuscript response

The submitted manuscript presents a review focused on factors influencing tablet disintegration and dissolution, particularly the relationship between tablet microstructure and its disintegration/dissolution, as well as methods used for characterization of these properties. The manuscript provides interesting insight into the literature findings related to this topic. However, the following points should be considered.

There are numerous omissions throughout the manuscript regarding the reference citing. i.e. the numbers enclosed in parentheses do not agree with the numbers of the corresponding references in the list. Please carefully check the reference list and all numbers in parentheses within manuscript text.

Thank you for the comment. The authors apologize for the problem in the reference listing and citing. The reference list and all the numbers in parentheses within the manuscript text have been corrected.

Lines 49-50. What is meant by compression strength? Tablet mechanical strength is usually characterized by the breaking force (“hardness” as a term is not preferred, please refer to USP), while compression pressure is important process parameter that can affect tablet mechanical properties (e.g. tablet breaking force, friability).

Thank you for this comment. In agreement with the reviewer’s comment, the word hardness has been replaced by tablet breaking force in line 51-52 (page 2).

Line 87. In order to more clearly distinct between liquid properties and characteristics of tablet, please correct this part as follows: liquid properties and tablet's microstructural properties.

Thank you for this comment. The part in now line 90 (page 2) has been corrected based on the reviewer’s recommendation.

Line 414-415, line 421. The meaning of the following phrases is not clear: "high normal load compacted tablets", "low normal loads". Please explain and correct. 

The phrases in lines 418 and 427 (page 10) have been corrected to “manufactured under high compaction pressure” and “low compaction pressure”, respectively.

Line 441-448. Immediate release preparations and rapid drug release are not typical only for orally disintegrating tablets, and therefore this paragraph might be misleading and should be corrected. There is a variety of different solid dosage forms showing immediate release of the API. Typical property of orally disintegrating tablets (orodispersible tablets) is rapid disintegration within the oral cavity in the presence of salivary fluid.

Thank you very much for this comment. In agreement with the reviewer’s comment this section has been clarified in lines 445-448 (page 13). Table 2 has also been added to exemplify products used as immediate release, extended, and delayed release.

Table 3. The parameter L/S ratio should be defined.

The parameter L/S ratio in Table 4 (previously Table 3) is now defined as liquid to solid (L/S) ratio. 

Table 4. The abbreviation FBG has not been introduced previously.

The abbreviation FBG has been corrected to fluid bed granulation in Table 5 (previously Table 4).

Table 4. The influence of binder droplet size in FBG is not clear. Please explain the influence of droplet size on wettability.

The influence of binder droplet size in fluid bed granulation has been explained in Table 5.

The following references are cited in Table 4, which presents the influence of process parameters for wet granulation in fluid bed granulator: Tan et al. Powder Tech. 2004, 143–144, 65–83; Chua et al. Chem. Eng. Sci. 2013, 98291-297. However, these studies are related to hot melt granulation that is considerably different process, with different granulation mechanisms involved and different granule morphology and microstructure.

The authors apologize for this confusion. These references were supposed to be removed from the text and the reference list. They are now removed from the Table and the reference list.

Caption for Table 4 should be corrected considering that only influence of process parameters was discussed not the influence of formulation parameters.

In agreement to the reviewer’s comment, caption for Tables 3-5 have been corrected to account only for process parameters.

Line 627. Mannitol can be filler not disintegrant. Please correct this.

Thank you for this comment. The word disintegrant was replaced by filler in line 649 (page 21).

Reviewer 3 Report

The manuscript entitled, ‘The Significance of Tablet Internal Structure on Disintegration 2
and Dissolution of Immediate Release Formulas: A Review’ discussed the tablet based formulations for drug delivery applications. The article nicely discussed but still I am mentioning some points which should be justified before publication;

1.      The author should address why tablets are better than other conventional; drug administration pathways.

2.      The release mechanism from tablets needs to be addressed also. Better to provide some schematics.

3.      Application areas are not discuss. Some specific examples should be given where tablet erosion are applied in clinical practices.

4.      Some articles have significance with drug release for your reference: https://doi.org/10.1021/acsabm.2c00664; https://doi.org/10.1016/j.jtemb.2022.127107; https://doi.org/10.1016/j.ijbiomac.2022.07.166.   

Author Response

Comment

Manuscript response

The manuscript entitled, ‘The Significance of Tablet Internal Structure on Disintegration 2
and Dissolution of Immediate Release Formulas: A Review’ discussed the tablet based formulations for drug delivery applications. The article nicely discussed but still I am mentioning some points which should be justified before publication;

1.      The author should address why tablets are better than other conventional; drug administration pathways.

Thank you for comment. A section in the introduction (lines 23-27) had already been included in the manuscript. However, to complement the review, which focus on immediate release tablets, a comparison between tablets and capsules is now added in lines 27-29 (page 1).

2.      The release mechanism from tablets needs to be addressed also. Better to provide some schematics.

4.      Some articles have significance with drug release for your reference: https://doi.org/10.1021/acsabm.2c00664; https://doi.org/10.1016/j.jtemb.2022.127107; https://doi.org/10.1016/j.ijbiomac.2022.07.166.  

Thank you very much for this comment. A schematics of active ingredient release in immediate release tablets has been added in Figure 3. This schematic is based on the study of immediate release tablets provided in reference 50. The authors appreciate the references provided by the reviewer. However, most of them are from extended or delayed systems which does not fall into the scope of this review on immediate release tablets.

3.      Application areas are not discuss. Some specific examples should be given where tablet erosion are applied in clinical practices.

Example of tablets along with their clinical practices are now provided in Table 2.

Round 2

Reviewer 2 Report

Reviewersʼ comments have been satisfactorily addressed and the quality of the manuscript has been improved. Therefore, I recommend the acceptance of the manuscript.

Reviewer 3 Report

The comments are not duly clarified by the author. The references mentioned by the reviewer are not highlighted in the text. This article can be published after completion of these.